# Phenotype "Explosion" in *Hercinothrips femoralis* (O. M. Reuter 1891) (Thysanoptera: Thripidae): A Particular Phenomenon for Successful Introduction of Economic Species

Rudolf Masarovič [1,*], Martina Zvaríková [1], Michaela Marcišová [1], Zuzana Ježová Provaznik [1], Pavol Prokop [1,2] and Peter Fedor [1]

1    Department of Environmental Ecology and Landscape Management, Faculty of Natural Sciences, Comenius University, Ilkovičova 6, 842 15 Bratislava, Slovakia; martina.zvarikova@uniba.sk (M.Z.); pavol.prokop@uniba.sk (P.P.); peter.fedor@uniba.sk (P.F.)

2    Institute of Zoology, Slovak Academy of Sciences, Dúbravská Cesta 9, 845 06 Bratislava, Slovakia

\*    Correspondence: rudolf.masarovic@uniba.sk

**Abstract:** Intraspecific trait variability, produced by genetic variation and phenotypic plasticity within species, allows the optimization of individual's fitness in different conditions, ultimately enhancing survival and reproduction. We investigated variability in morphological traits of invasive thrips species *Hercinothrips femoralis* (O. M. Reuter, 1891) during simulated introduction and establishment in a novel environment. Six generations of this species were reared in laboratory for eight months. The initial phase of introduction was simulated by the transfer of thrips generations to environments with different environmental conditions varying in temperature and humidity. The statistical evaluation of seven measured morphological attributes (e.g., body length, wing length) was performed to analyse the morphological variability. Species phenotypic "explosion" in several morphological characters (especially total body length) was observed during the initial phase of introduction in generations brought from the primary site into novel environments with different conditions. Probable phenotypic specialization was observed during the generations following introduction under the same ecological conditions. Furthermore, the most variable morphological features were specified. This study goes beyond the taxonomic level, because its results and main idea can be applied to any invasive species.

**Keywords:** phenotype "explosion"; biological invasion; morphological variability; intraspecific variability

## 1. Introduction

Invasive species have become a serious problem in Europe, with significant consequences for natural ecosystems as well as urban and farmland areas, often as a synergy of climate change and the globalisation of biological commodity trade [1]. The study of these species' establishment in a new geographic area under the specific local conditions represents the first step towards developing an effective pest control strategy [2]. Therefore, studying the mechanisms that lead to their survival and successful establishment in novel environments is of great importance in recent decades.

In this regard, the phenotype and its variability are crucial phenomena in the biology and distribution of invasive species. It helps us to understand the adaptations of individual specimens through phenotypic plasticity as a tool by which an individual adapts to the changing environment [3]. Every species is characterised by intraspecific trait variability (ITV), which is necessary in the adaptation of individuals or species to environmental stresses [4,5]. ITV can be produced by genetic variation and phenotypic plasticity [6].

Genetic variation within a population is characterized by individuals with diverse genetic information (distinct genotypes) produced by different evolutionary processes. These specimens have a tendency to produce their own characteristic phenotypes [6,7]. The local environment selects the phenotypes with the most suitable traits, and these

individuals have a higher chance to survive and improve their fitness. The phenotype is heritable and is more probable to appear in the offspring [6].

On the contrary, phenotypic plasticity can be defined as the ability of an organism (its genotype) to react to an environment with a change in form, state, movement or rate of activity [8] with the production of different phenotypes [9]. Any abiotic or biotic factor can result in plastic responses and may lead to highly integrated and environmentally sensitive production of alternative phenotypes by a given genotype under various environmental conditions [10].

Developmental plasticity, the ability of organisms to alter their phenotype during the embryonic or larval stage based on environmental factors [11], can enhance establishment success of the introduced organisms in a novel environment, support their invasive potential, decrease vulnerability against environmental changes, and reduce the risk of extinction [12,13]. Genetically identical individuals whose development took place in different conditions can differ in many morphological, physiological, or behavioural characteristics [14–16]. Each individual faces a slightly different environment during its development, causing plasticity leading to ITV [6]. Increased ITV as a result of species plasticity leads to increased population fitness, and the most variable populations perform better in general [5,6].

Research on phenotypic and developmental plasticity has intensified in recent decades, with interest in finding out how intraspecific variability and plasticity affect the performance and success of individuals and populations [17]. Invasive organisms probably display higher phenotypic plasticity than their non-invasive counterparts [18], and continuous invasions to new environments often point towards a more phenotypically plastic species [19]. Under these findings, previous studies indicate that phenotypic plasticity plays an important role in the success of invasive organisms [20].

Based on previous information, the probability of successful establishment in novel environments increases with the genotypic and phenotypic diversity in founder groups of invasive organisms [17]. In this sense, if a species occurs for a long time in some environment, its ontogeny is likely to produce specialized forms. When conditions change (invaded environment) there may be a shift to more generalized forms with a significant degree of variability in functionally important traits (phenotype "explosion") during the initial phase of introduction in the first generations (founder groups) to better fit into a given environment. Subsequently, it probably searches for the most suitable phenotype (phenotype specialization) for those specific conditions. We refer to this phenomenon as "explosion" and the subsequent specialization of the phenotype.

*Hercinothrips femoralis* (O. M. Reuter) was used as a model of invasive species, in which the morphological changes during the initial phase of the introduction were studied. The banded greenhouse thrip is an important polyphagous pest species with cosmopolitan distribution [21,22] and may cause significant economic damage [23]. It represents a glasshouse phytopathogenic species in temperate regions [24] that is successfully moving from greenhouses into private households [25]. In addition, this species can also spread outside greenhouses into previously inaccessible, non-hostile environments [26] with specialized dispersion mechanisms [27,28]. Moreover, its introduction to a new environment often goes unnoticed, and the species is usually discovered after it has already established itself under novel conditions [23]. *H. femoralis* represents a typical haplodiploid thrips species. Males are haploid and females diploid capable of parthenogenesis. *H. femoralis* usually reproduces by thelytoky [27,29]. Females represent the dominant part of offspring, and males are very rare. There are two forms of thelytokous modes of reproduction. It can be encoded in genomes as a non-reversible mechanism or microbe-induced by maternally inherited bacteria (such as Wolbachia) characterized by reversibility (for example, through treatment with antibiotics) [30,31]. The reversible microbe-associated thelytokous mode of reproduction is usually induced by maternally inherited bacteria such as Wolbachia (this is often the case with *H. femoralis*). An *H. femoralis* population treated by antibiotics may produce males that copulate with females [30].

Our goal was to test the phenotypic response and variability of this species as the main factors of a successful establishment in a novel environment (laboratory controlled). We hypothesized that invasive species suddenly increase their morphological variability in the first introduced generations (phenotype "explosion"–generalized forms) and try to find a suitable phenotype in new conditions through the next generations ("specialisation" of phenotype–specialized forms). Undisputedly, this is a unique adaptation mechanism complex, which guarantees high fitness of a species along ecological gradients as a predisposition of alternative ontogenies for potentially more successful infiltration into new (secondary) ecological niches, as well as expansion within the distribution area.

## 2. Materials and Methods

The changes in the phenotype and its "explosion" were studied in model species *H. femoralis*. Individual generations of *H. femoralis* were reared in breeding boxes (Figure 1) with dimensions of 26 × 19 × 14 cm. Openings were cut in the sides, covered with nets, and properly fixed with double-sided tape. A light was installed above the box, and the temperature was recorded by a thermometer (Comet Logger s3120). Cucumbers were used as food for thrips [32]. All fruits were stored in the same conditions and came from one distributor before the experiment. They were regularly checked during the experiment to prevent them from rotting, and food of good quality was offered to each generation.

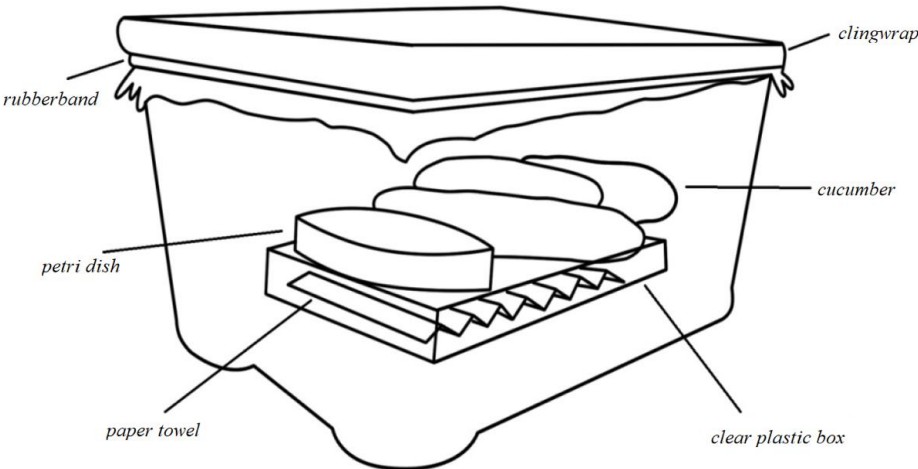

**Figure 1.** Rearing box (modified from Štefánik 2020 [32]).

### 2.1. Experimental Design

The population of *H. femoralis* was representative, reared for many generations in a greenhouse in closed laboratory conditions. Generation G0 represented the third generation reared in a greenhouse in "stable conditions" (ST—stable temperature and humidity), in which temperature (27 ± 1 °C) and humidity (75% ± 10%) did not change. The other three generations (G1, G2, G3) were moved outside the greenhouse to "unstable conditions" (NS), in which the temperature varied between 19.1 and 25 °C (average 22.2 °C) and humidity between 22.4 and 41.8% (average 31.9%) depending on the surrounding environment. In the second part of the experiment, the generations (G4, G5) were moved to the greenhouse again, where a stable temperature (23 °C ± 0.5 °C) and humidity (75% ± 10%) were maintained. The movements of thrips and their development within three different environments (E1 (ST)—G0; E2 (NS)—G1, G2, G3; and E3 (ST)—G4, G5) were supposed to simulate the process of introduction of thrips into a new environment with different conditions. Type and amount of food, space– (i.e., breeding boxes), type of lighting, photoperiod, etc., were equal and controlled (Figure 2).

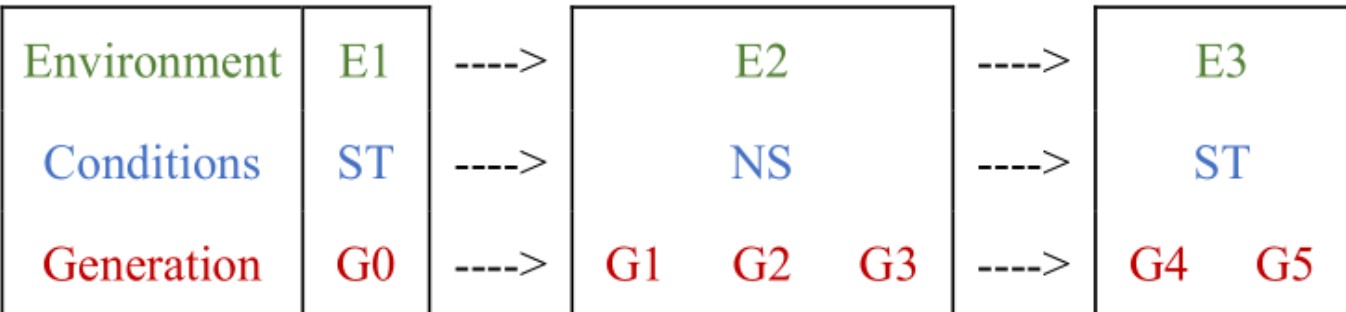

**Figure 2.** Experimental design and rearing of 6 consecutive generations (G0 to G5) in three types of environments (E1, E2, E3) in stable (ST) and unstable (NS) climatic conditions. Generations G1 and G4 were "introduced" to the new environment.

At the beginning, only the adult individuals from the generation G0 reared in a greenhouse in stable conditions were relocated to different breeding boxes that were placed in "unstable conditions" (E2 environment). Breeding boxes (Figure 1) were prepared as mentioned before with an equal number of cucumbers that were washed with sodium hypochlorite (NaClO, 400 ppm) and subsequently washed with water and dried. The adults of the previous G0 generation were taken away when the larvae of the new G1 generation began to appear. When adults of the G1 generation were present, they were moved to a new breeding box, and the process was repeated. This procedure was repeated with each generation, and adults were collected and preserved in AGA solution (60% ethyl alcohol, glycerol, acetic acid) and used for subsequent morphometric analyses.

### 2.2. Selection and Analysis of Morphological Variables of the Species Hercinothrips Femoralis

The samples for morphometric analyses of phenotypes were taken by measure of individuals from six generations (388 individuals of *H. femoralis*), which were reared for 8 months. Of the total number, 298 individuals (specifically 48 specimens in G0, 23 in G1, 61 in G2, 54 in G3, 73 in G4, 39 in G%) and only females were measured. A total of 2086 measurements were performed. Sex ratio of females to males was 375:12. Males and damaged, not well-positioned individuals were excluded from the measurements.

The length of individual development cycles lasted an average of 33.6 days. Seven morphometric characteristics were measured on each individual: head width (HW), pronotum length (PL), pronotum width (PW), forewings length (WL), clavus length (CL), ovipositor width (OW), and body length (BL) (Figure 3).

Morphometric characteristics were selected according to the works of Fedor et al., 2008 [33], Fedor et al., 2014 [34], and Zvaríková et al., 2016 [35], and were measured using a LEICA DM 2000 LED microscope and a camera with LAS EZ software with a 10-fold (10×) eyepiece. The mounting process and all measurements were performed by one person. Only undamaged individuals were used during the process of measurement to avoid measurement errors or missing data. Broken or extremely twisted individuals were excluded.

### 2.3. Statistical Analysis

The changes in variability between individual generations were indicated by standard deviation (*SD*), variance ($s^2$), and interquartile range (*IQR*) of measured quantitative variables. Levene's test was used to analyse non-equality of variances and evaluate significant changes in homogeneity, respectively, non-homogeneity of variances between individual pairs of generations (e.g., G0 and G1) that followed one after the other. This test was calculated separately for each pair of generations, while significant changes indicate "explosion" of the phenotype during the primary phase of introduction and subsequent specialisation. A Principal Components Analysis (PCA) with 75% confidence ellipses ruled out the potential outliers and summarized variation in measured data. Moreover, PCA diagram

with confidence ellipses implied the changes in variability between individual generations. Statistical analyses and creation of plots were performed in R-software version 4.2.1. [36]. The "car", "devtools", "ggplot2/ggbiplot", and "lattice" packages were used to perform analyses and create graphs.

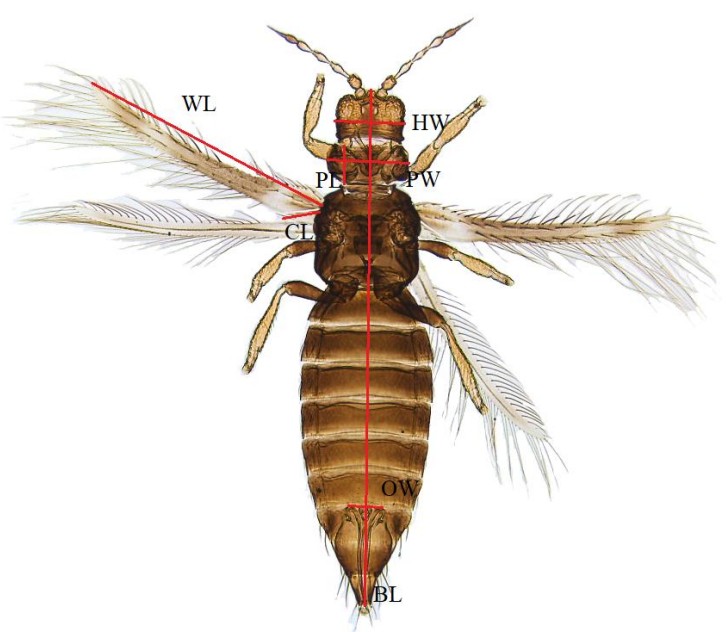

**Figure 3.** Measured morphological parameters: head width (HW); pronotum width (PW); pronotum length (PL); total body length (BL); clavus length (CL); forewings length (WL); ovipositor width (OW).

## 3. Results

### 3.1. Phenotype "Explosion" during Primary Phase of Introduction

The standard deviation of the measured morphological characteristics within individual generations changes when the environment has changed ("introduction phase", generations G1 and G4) (Table 1). The idea of an "explosion" of the phenotype is best seen with the values of the total body length (BL), pronotum length (PL), clavus length (CL), and wing length (WL).

**Table 1.** Standard deviations (*SD*) of the measured morphological characters within individual generations.

| *SD* | HW | PL | PW | WL | CL | OW | BL |
|------|------|------|------|-------|------|------|--------|
| G0 | 5.41 | 7.25 | 8.11 | 36.68 | 5.94 | 5.02 | 75.33 |
| G1 | 4.58 | 8.25 | 7.92 | 38.91 | 6.01 | 5.77 | 116.01 |
| G2 | 5.01 | 8.06 | 8.16 | 37.40 | 5.87 | 6.23 | 97.80 |
| G3 | 4.09 | 5.70 | 7.14 | 33.09 | 4.35 | 5.08 | 84.62 |
| G4 | 3.70 | 9.77 | 6.83 | 39.39 | 5.15 | 5.24 | 121.42 |
| G5 | 3.73 | 7.43 | 6.75 | 35.87 | 3.94 | 7.07 | 64.13 |

HW, PL, PW, WL, CL, OW, BL—measured morphological features, see the Materials and Methods section; G0, G1, G2, G3, G4, G5—generations of *H. femoralis* within individual environments, see Materials and Methods. For better interpretation, the colours indicate changes in morphometric variability through the process of phenotypic "explosion" and specialisation: dark blue—highest value = "explosion" of the phenotype; paler blue—medium value; light blue—lowest value = "specialization".

In this regard, the standard deviation of total body length (BL) in the G0 generation under E1 conditions was 75.33. By changing the conditions (from environment E1 to environment E2—"introduction") there was an "explosion" of the phenotype in G1 with an increase in variability in body length (s = 116.01—an increase in variability by 35.1%

compared to G0). Subsequently, in the process of specialization, the value of the standard deviation decreased in the generation G2 (s = 97.80—a 15.7% decrease compared to G1) and in the G3 generation (s = 84.62—decreased by 13.5% compared to G2). By changing conditions (from the E2 environment to the E3 environment), an "explosion" occurred again in the G4 generation phenotype (s = 121.42—a 30.3% increase compared to G3). In the next generation, the value of standard deviation of body length dropped to 64.13 (a 47.18% decrease compared to G4).

The values of variances ($s^2$) in the measured morphological features within individual generations are shown in Table 2. The values for total body length (BL), pronotum length (PL), clavus length (CL), and wing length (WL) (as standard deviation values) showed the most visible changes in terms of the idea of an increase in phenotype variability in the first phase of introduction. In the conditions of E1 (G0), the BL variance value was 5674.08. The variance in the phenotype changed significantly ($p < 0.01$, see Table 3) when the environment changed. In the first generation (G1), the variance increased to 13,458.03 (an increase of 57.84% compared to G0) and gradually decreased in the second generation G2 to 9565.71 (a decrease of 28.92% compared to G1) to a value of 7159.71 (a decrease of 25.15% compared to G2) in the third generation G3. Again, by changing the conditions (E3 environment) the variance in the fourth generation G4 increased significantly ($p < 0.05$) to 14,742.25 (an increase of 51.43% compared to G3) and subsequently decreased in the fifth generation G5 ($p < 0.01$, see Table 3) to 4112.21 (reduced by 72.11% compared to G4).

**Table 2.** Variances ($s^2$) in measured morphological characters within individual generations.

| $s^2$ | HW | PL | PW | WL | CL | OW | BL |
|---|---|---|---|---|---|---|---|
| G0 | 29.28 | 52.63 | 65.79 | 1345.42 | 35.33 | 25.25 | 5674.08 |
| G1 | 21.00 | 68.07 | 62.80 | 1514.05 | 36.16 | 33.31 | 13,458.03 |
| G2 | 25.06 | 64.98 | 66.50 | 1398.66 | 34.46 | 38.81 | 9565.71 |
| G3 | 16.73 | 32.51 | 50.96 | 1095.11 | 18.96 | 25.77 | 7159.71 |
| G4 | 13.70 | 95.49 | 46.69 | 1551.62 | 26.51 | 27.49 | 14,742.25 |
| G5 | 13.95 | 55.26 | 45.53 | 1286.63 | 15.56 | 49.99 | 4112.21 |

HW, PL, PW, WL, CL, OW, BL—measured morphological features, see the Materials and Methods; G0, G1, G2, G3, G4, G5—generations of *H. femoralis* within individual environments, see Materials and Methods. The colours indicate changes in morphometric variability through the process of phenotypic "explosion" and specialisation: dark blue—highest value = "explosion" of the phenotype; paler blue—medium value; light blue—lowest value = "specialization".

**Table 3.** Testing the homogeneity of variances (Leven's test) between morphometric traits of individual generations indicating an "explosion" (between G1 and G0 and G4 and G3) or possible specialization (between G2 and G1, G3 and G2, and G5 and G4) of the phenotype.

| | HW | PL | PW | WL | CL | OW | BL |
|---|---|---|---|---|---|---|---|
| G1-G0 | | | | | | | ** |
| G2-G1 | | | | | | | |
| G3-G2 | . | . | | | * | | |
| G4-G3 | | *** | | | | | * |
| G5-G4 | | * | | | . | | ** |

Signif. codes: 0 '***' 0.001 '**' 0.01 '*' 0.05 '.' 0.1 ' ' 1

HW, PL, PW, WL, CL, OW, BL—measured morphological features, see the Materials and Methods; G0, G1, G2, G3, G4, G5—generations of *H. femoralis* within individual environments, see Materials and Methods.

Similar results were observed in the interquartile range (*IQR*), in which the idea of "explosion" and subsequent specialization of the phenotype was best expressed in the morphometrics of total body length, pronotum length, and wing length (Table 4).

**Table 4.** Interquartile range (*IQR*) of measured morphological features within individual generations.

| *IQR* | HW | PL | PW | WL | CL | OW | BL |
|---|---|---|---|---|---|---|---|
| G0 | 7.83 | 11.54 | 11.19 | 45.35 | 9.57 | 6.67 | 95.39 |
| G1 | 4.74 | 10.23 | 12.85 | 53.88 | 8.60 | 7.66 | 190.05 |
| G2 | 5.90 | 9.18 | 9.40 | 54.05 | 9.02 | 9.82 | 149.41 |
| G3 | 4.44 | 8.04 | 10.79 | 40.80 | 4.54 | 5.85 | 94.71 |
| G4 | 5.79 | 17.22 | 9.14 | 55.53 | 7.55 | 6.74 | 177.59 |
| G5 | 5.58 | 9.91 | 7.87 | 44.22 | 5.44 | 11.51 | 89.45 |

HW, PL, PW, WL, CL, OW, BL—measured morphological features, see the Materials and Methods; G0, G1, G2, G3, G4, G5—generations of *H. femoralis* within individual environments, see Materials and Methods. The colours indicate changes in morphometric variability through the process of phenotypic "explosion" and specialisation: dark blue—highest value = "explosion" of the phenotype; paler blue—medium value; light blue—lowest value = "specialization".

The measured values of the most significant morphological traits and their distribution are indicated in box plots (Figure 4), where the variability in the selected morphological characters is changing because of the changing environment and individual generations, which may indicate the "explosion" and specialization of these traits.

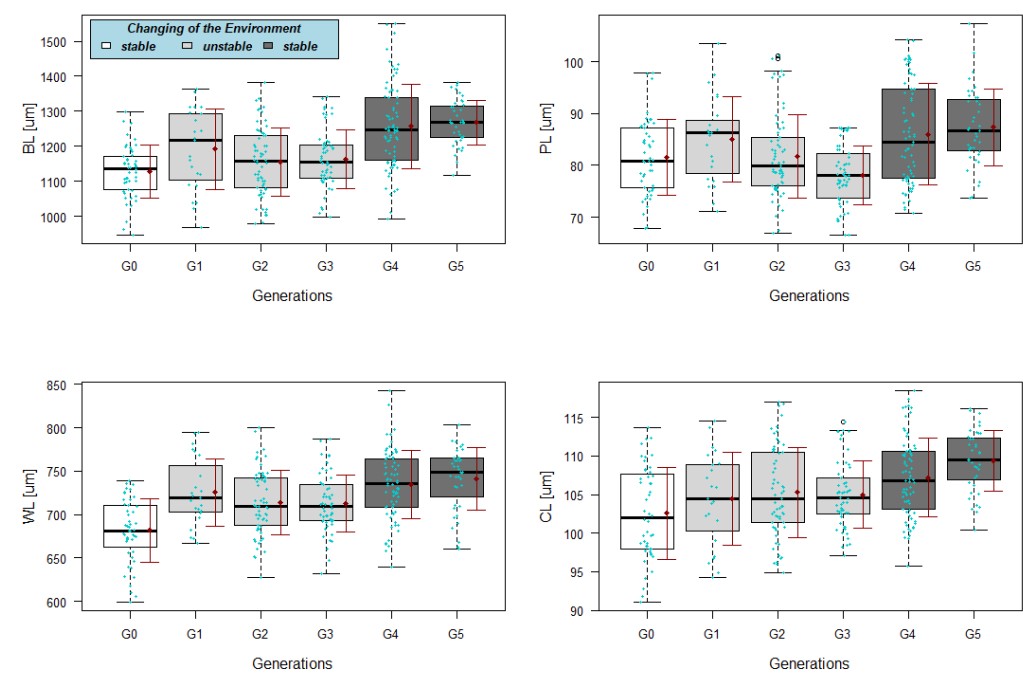

**Figure 4.** Quantitative distribution of the most significant measurements of total body length (BL), pronotum length (PL), wing length (WL), and clavus length (CL). Blue dots—individual measurements; red points and arrows—mean and standard deviation.

### 3.2. From Generalization (Phenotype "Explosion") to Specialization (Selection of a Suitable Phenotype)

The monitoring of the selection within three (G1–G3) or two generations (G4–G5) is indicated by the graphs in Figures 5 and 6, which imply some interesting tendencies in phenotype selection in each environment.

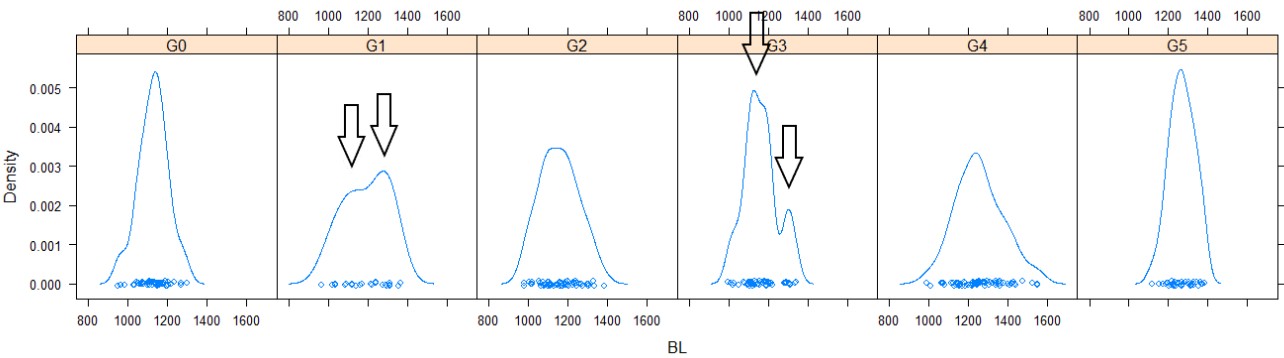

**Figure 5.** Indication of specialization of total body length (BL) expressed by density (density plot).

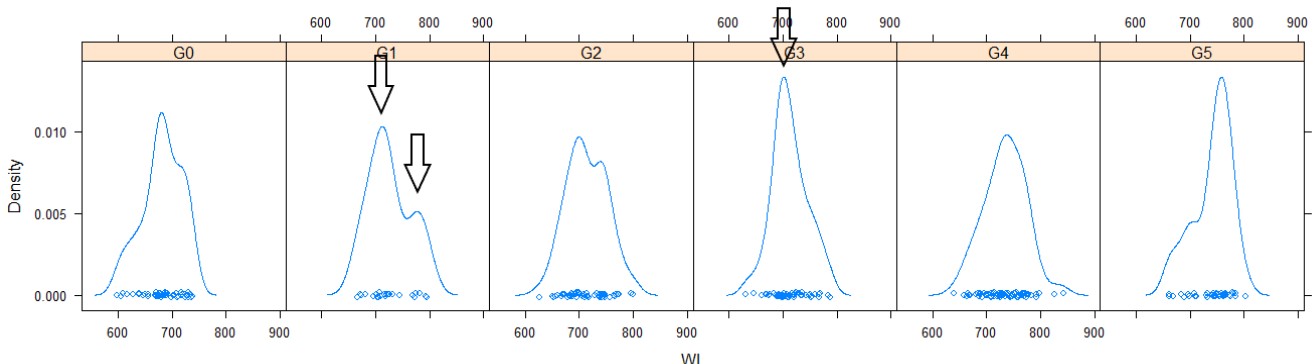

**Figure 6.** Indication of selection of wing length (WL) expressed by density plot (density plot).

Figure 5 indicates the specialization of total body length (BL) within two different environmental conditions (temperature change). Two forms of body length appear in generation G3 (the last generation in environment E2), which implies the disruptive type of phenotype specialization. It appears that *H. femoralis* produces two forms, a first form (more numerous) with a smaller body size and a second, bigger form. Moreover, the density plot indicates that *H. femoralis* "creates" only one form in the third environment (E3) with the stable temperature in the G5 generation.

On the contrary, the stabilizing type of specialization is indicated in Figure 6 in the E2 environment, where only one form of wing length (WL) in *H. femoralis* in generation G3 (the last generation in the E2 environment) is produced. Moreover, the density plot indicates that *H. femoralis* also "creates" only one form of wing length in the third environment with stable temperature in generation G5 (environment E3).

The Principal Components Analysis (PCA) emphasized more than 75% confidence ellipses to rule out the potential outliers. The results of the PCA are summarised in the ordination diagram in Figure 7. The first two axes (shown in the diagram) explain 86.2% of the total variation in the measured data. The ordination diagram indicates that G1 and G4 show more variability compared to G0, G2, G3, and G5. According to the angles between the arrows of variables, the total body length positively correlates with pronotum length and wing length positively correlates with clavus length.

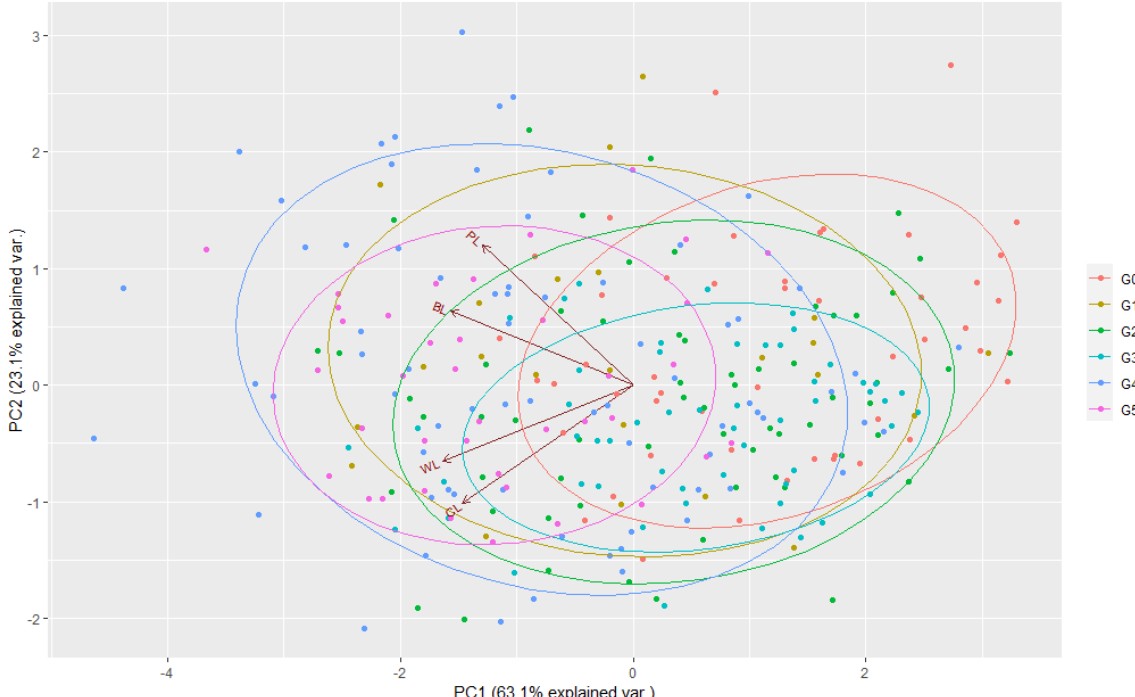

**Figure 7.** The ordination diagram of the Principal Components Analysis (PCA) of G0–G5 with selected morphological characteristics BL, WL, CL, and PL. BL—total body length; CL—clavus length; WL—wing length (details in Materials and Methods); G0, G1, G2, G3, G4, G5—generations of *H. femoralis* within individual environments (details in Materials and Methods).

## 4. Discussion

Our results indicate that invasive and introduced species have a tendency to phenotypically adapt in novel environments, most probably through mechanisms of phenotypic plasticity to find suitable phenotypes in new conditions with the aim to successfully establish a new population and spread in the future.

An "explosion" of a phenotype during the initial phase of introduction of *H. femoralis* to a novel environment was recorded in the first generation depending on the selected morphological trait. The transfer of thrips specimens to different environmental conditions (temperature + humidity) caused a significant change in the variability in specific characteristics (e.g., total body length) during the next generation. It is likely that by creating multiple body forms (bigger body/smaller body), the invasive species has a "choice" of which phenotype is best for the environment (selection may occur) and has a higher chance of establishing and succeeding [37,38].

As expected, some traits showed a higher degree of variability. The greatest trend in the sense of the "explosion" of the phenotype within the first generation during primary phase of introduction was significantly observed in the total body length (BL) and by using measures of variability ($SD$, $s^2$, $IQR$) indicated in wing length (WL), clavus length (CL), and pronotum length (PL). Our results are similar to the work of Fedor et al. [39], where the most variable characteristics were total body length, pronotum width, wing length, and ovipositor length. Some characteristics are probably more variable compared to others [35]. These conclusions indicate the functional importance of some "plastic" characteristics over others [6] and that plasticity in body size allows animals to survive even in variable conditions [40]. Moreover, the greater phenotypic diversity of functionally important traits increases the success of settling and distribution to a new environment [17,41].

For example, body size is one of the most remarkable and significant characteristics of all organisms. Strong relationships between body size and various physiological and ecological traits exist, including metabolic rate, production rate, survival probability,

fertility, and distribution [42,43]. In this regard, over the course of several generations, specialization of morphological traits may gradually occur. In our case, the total body length was characterized by two forms ("disruptive" specialization) in generations G1 and G3 in the E2 environment. This can be related to intraspecific competition or successful invasion in dispersal ability and establishment of new generations. In this sense, species with larger bodies may have an advantage in the competition and reproductive success [44]. On the contrary, species with a smaller body are favoured in search narrow spaces (thigmotaxis) that provide them with shelter and protection against inhospitable conditions and predators [29,45,46]. Moreover, body size depends on several factors and conditions in the environment occupied by the species [8,47], and its variability may gradually decrease in environments with stabile conditions. We could also find similar parallels with other morphological characteristics (WL, PL, CL).

Furthermore, the ability of introduced species to invade novel habitats may depend largely on dispersal ability [48,49] when exotic species may display a higher frequency of dispersal (higher dispersal ability) vs. non-dispersal (lower dispersal ability) of functional phenotypes [6]. This may be associated with increased variability in the total body length [50], forewings length [51], or clavus length of *H. femoralis* in our research during the initial phase of introduction to a novel environment and to its "search" for a suitable specialized phenotype. For example, the trade-off and selection of dispersive or non-dispersive individuals with different total body lengths (in generation G3) may be related to the dispersal ability in novel environments [50,52,53]. Variability in and selection of wing length, clavus length, and body length thus can be functionally related to migration mechanisms [54,55].

To prevent invasive species from colonizing and spreading outside their original area of occurrence, it is important to monitor the factors and mechanisms that increase their success [56–58]. There are several mechanisms by which phenotypic diversity can positively influence the emergence and persistence of invasive species in the environment [5,59]. These mechanisms include a higher the likelihood that more diversified groups have preadapted phenotypes [16,60]. Recognizing the role of genotypic and phenotypic diversity in invasive species may improve our ability to prevent potential harmful invading species from spreading outside of their native areas [17], and approaches aimed at important traits of invasive species may help to implement management for their effective control. The most plastic characteristics can help to identify and predict species that can be a potential threat to biodiversity and agriculture. The identification of potentially dangerous species in the primary phase of introduction can be one of the effective mechanisms to prevent major damage and a prompt solution for plant protection. Our analyses have emphasized the challenge of specific phenotypic "explosion" curves in forensic reconstruction of the banded greenhouse thrip *H. femoralis* (Thysanoptera: Thripidae: Panchaetothripinae) in terms of introduction timing and generation assessment, with an outstanding postulation for applied phytosanitary care.

## 5. Conclusions

The main idea of this study is applicable to any invasive economic pest species from different taxonomic levels, and more studies need to be conducted on different invasive and quarantine species. Increased variability in morphological traits of the pest species *H. femoralis* during a simulated introduction in a novel environment was examined. Species phenotypic "explosion", especially in total body length, but also indicated in wing length, pronotum length, and clavus length, was observed in the first generations during the initial phase of biological invasion. Probable phenotypic specialization was observed during the generations after introduction under the same ecological conditions. How such specializations increase individuals' fitness needs to be addressed in future research.

**Author Contributions:** All authors contributed to the study conception and design. Material preparation and data collection: R.M., M.M. and Z.J.P.; Data curation: R.M. and M.Z.; Methodology: R.M. and M.Z.; Data analysis: R.M.; Writing—original draft preparation: R.M.; Writing—review and editing: M.Z., P.F. and P.P.; Formal analysis and investigation: R.M. and Z.J.P.; Visualization: R.M. and Z.J.P.; Funding acquisition: P.F. and P.P.; Resources: P.F. and P.P.; Supervision: P.F. and P.P. All authors have read and agreed to the published version of the manuscript.

**Funding:** Supported by the Scientific Grant Agency of the Ministry of Education of the Slovak Republic, Grant No. VEGA 1/0286/20.

**Data Availability Statement:** The data presented in the study are available on request from corresponding author.

**Conflicts of Interest:** The authors declare no conflict of interest.

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
