# Peer review of "Phenotype “Explosion” in Hercinothrips femoralis (O. M. Reuter 1891) (Thysanoptera: Thripidae): A Particular Phenomenon for Successful Introduction of Economic Species"

_horticulturae, doi:10.3390/horticulturae9121327_

Round 1
Reviewer 1 Report
Comments and Suggestions for Authors
This is a well-written manuscript. The language is fluent and it makes the ms a pleasure to read. However, I would like to call attention to a significant shortcoming. The authors report significant variance in pronotum and total body length and it is attributed as phenotypic plasticity in response to varying environmental conditions between the sampled consecutive generations of Hercinothrips femoralis. However, it is not clear in the current form of the ms, why genetic diversity is excluded from the potential sources of the observed variance. The aim of this paper is to report phenotypic plasticity in the studied model organism, therefore I expect more emphasis on how genetic diversity is minimized in the studied thrips population. More details should be reported about this in the materials and methods sections and potentially in the introduction section as well. As one of the measured characters is total body length and this particular variable is very sensitive to the circumstances of slide mounting, I expect more details here too, reporting all the efforts that were made to reduce measurement variance (standardisation of slide mounting and measurements).
Hercinothrips femoralis reproduces by endosymbiont mediated thelytoky and therefore only females are usually present in populations, males are very rare. It makes this thrips species a good model organism to study phenotypic plasticity. Isofemale lines are easily created by isolating individual females whose progeny might be considered as individuals with identical genome. This is the basis of any study aiming to measure phenotypic plasticity. However, no details are provided about this in the current form of the ms. The reproductive biology of the studied thrips species is not introduced, so those readers who are not familiar with it, might not understand why this model organism is selected for this study. I suggest adding information about this in the introduction section.
A model organism with apomictic thelytokous reproductive mode would have been ideal for creating isofemale lines with identical genome (completely excluding genetic diversity from the studied population). What is known about the genetic diversity of Hercinothrips femoralis populations that reproduce by endosymbiont induced thelytoky? Is it central fusion or terminal fusion leading to female production in this species? More details should be provided about this and its consequences on the genetic diversity of the studied population.
Were there any males observed in this study? In line 149 it is stated that only females were measured but does it also mean that there were no males present or some males were present but they were excluded from the measurements?
Nothing is reported in the ms about the origin of the Hercinothrips femoralis population. It is only stated that the third generation of the population was first sampled under the first environmental condition set-up. How many Hercinothrips femoralis specimens were used to establish the first generation? If it was a single specimen used to establish the first generation, then all the consequent generations are the progeny of a single isofemale line. If more than one specimen were used to establish the original population, genetic diversity can not be completely excluded. One should not expect great genetic diversity in an adventive population of an invasive species, if the population size is small but genetic diversity can only be completely excluded in thelytokous isofemale lines (non-deleterious mutations being very unlikely in such a short period of time of 5 generations).
Author Response
Dear reviewer,
thank you very much for your comments and corrections. We appreciate that and considered intensively all the notes and suggestion you gave us. They indisputably improve the study. Please see the attachment.

Reviewer 2 Report
Comments and Suggestions for Authors
The article, which deals with the influence of the environment on the change in the phenotype of the selected species of thrips, Hercinothrips femoralis, is interesting for the fields of fundamental and applied zoology and represents a novelty in the field of research on interactions between the environment and animals (insects).
Using the appropriate methods, procedures and analyses, the authors studied the phenotypic response of the thrips species in several successive generations and obtained extremely interesting results, which will most likely be useful for other invasive animal species as well.
I suggest the acceptation of the paper for publication after minor revisions, as follows:
line 38: I suggest to mention the most important invasive alien insect species in Europe at the moment - Halyomorpha halys. I suggest to cite a paper, where this species is mentioned, for example: ROT et al. 2022. Biological parameters, phenology and temperature requirements of Halyomorpha halys (Hemiptera: Pentatomidae) in the Sub-Mediterranean climate of Western Slovenia. Insects, 13, no. 10, art. 956, 15 p.
line 78: when an organism is first mentioned in the text it should be written with a full latin name, i.e.: Hercinothrips femoralis (O. M. Reuter)
line 99: Did the H. femoralis specimens you used in the experiment come from a laboratory population? If so, which laboratory do they originate from and how long is the population grown in it?
Author Response
Dear reviewer,
thank you very much for your comments and corrections. We appreciate all the comments and suggestion you gave us. They indisputably improve the study. Please find our reactions in attached document.
